# DNA Hypomethylation in the TNF-Alpha Gene Predicts Rheumatoid Arthritis Classification in Patients with Early Inflammatory Symptoms

**DOI:** 10.3390/cells12192376

**Published:** 2023-09-28

**Authors:** Rujiraporn Pitaksalee, Rekha Parmar, Richard Hodgett, Paul Emery, Frederique Ponchel

**Affiliations:** 1Leeds Institute of Rheumatic and Musculoskeletal Medicine, University of Leeds, Leeds LS2 9JT, UK; 2Leeds University Business School, University of Leeds, Leeds LS2 9JT, UK; 3NIHR Leeds Biomedical Research Centre, Leeds Teaching Hospitals NHS Trust, Leeds LS1 9LF, UK

**Keywords:** DNA-methylation, qMSP biomarker, rheumatoid arthritis

## Abstract

Biomarkers for the classification of rheumatoid arthritis (RA), and particularly for anti-citrullinated peptide antibody (ACPA)-negative patients, remain an important hurdle for the early initiation of treatment. Taking advantage of DNA-methylation patterns specific to early RA, quantitative methylation-specific qPCR (qMSP) offers a robust technology for the development of biomarkers. We developed assays and established their value as RA classification biomarkers. Methods: DNA-methylation data were screened to select candidate CpGs to design qMSP assays. Eight assays were developed and tested on two early inflammatory arthritis cohorts. Logistic regression and bootstrapping were used to demonstrate the added value of the qMSP assays. Result: Differentially methylated CpG data were screened for candidate CpG, thereby meeting the qMSP assay requirements. The top CpG candidate was in the TNF gene, for which we successfully developed a qMSP assay. Significantly lower DNA-methylation levels were observed in RA (*p* < 4 × 10^−9^), with a high predictive value (OR < 0.54/AUC < 0.198) in both cohorts (n = 127/n = 157). Regression using both datasets showed improved accuracy = 87.7% and AUC = 0.944 over the model using only clinical variables (accuracy = 85.2%, AUC = 0.917). Similar data were obtained in ACPA-negative patients (n = 167, accuracy = 82.6%, AUC = 0.930) compared to the clinical variable model (accuracy = 79.5%, AUC = 0.892). Bootstrapping using 2000 datasets confirmed that the AUCs for the clinical+TNF-qMSP model had significant added value in both analyses. Conclusion: The qMSP technology is robust and can successfully be developed with a high specificity of the TNF qMSP assay for RA in patients with early inflammatory arthritis. It should assist classification in ACPA-negative patients, providing a means of reducing time to diagnosis and treatment.

## 1. Introduction

Suboptimal management of rheumatoid arthritis (RA) leads to irreversible joint destruction, function, and quality-of-life loss. It is widely accepted that effective treatment in the early stages can be the most beneficial [1,2,3,4,5]. Early access to treatment, however, requires early diagnosis. Current RA classification criteria [6] differentiate RA from other inflammatory arthritides (IA), with an overall sensitivity of 80–85% [7,8]. Classification heavily relies on the presence of autoantibodies, while only 50–60% of patients with early IA are antibody positive, at their presentation [9,10,11]. This highlights an important unmet clinical need for the development of additional biomarkers for early RA classification, especially for antibody-negative RA.

Epigenetic modifications are heritable, while dynamic changes do not alter the genetic code and result from environmental/external pressures and/or diseases. Epigenetic biomarkers have therefore been defined as epigenetic marks that can be measured and possess clinical utility. Epigenetic biomarkers have received great interest, recently [12] and are already in use in cancer, psychiatric, and neurodegenerative disorders, in particular the DNA-methylation biomarkers [13,14,15,16,17,18,19].

In RA, many alterations in DNA-methylation epigenetic profiles have been demonstrated [20,21,22,23,24,25,26,27,28,29,30,31,32,33], as well as histone modifications [34,35,36,37,38]. We recently described a series of differential DNA methylations in naïve CD4+ T-cells in patients with early RA [39]. This work provides a rationale for the selection of candidate CpGs to design assays quantifying DNA methylation. We chose the methylation-specific quantitative-PCR (qMSP) technology to develop novel biomarkers for the classification of RA.

qMSP assays target a specific change in DNA methylation at a particular CpG dinucleotide. This technology is highly dependent on the sequence to be targeted and the source of DNA, as contamination from various cell types causes dilution of the signal, and the amplitude of the differential methylation between groups or outcomes of interest presents further challenges in design criteria.

In this study, we screened our DNA-methylation data [39] and publicly available datasets to select candidate CpGs to design qMSP assays that could discriminate RA from other forms of IA. We used the assays to analyse patients with early IA symptoms from 2 cohorts who had either progressed or not to RA. We demonstrated the value of this assay as a classification biomarker, notably in ACPA-negative participants.

## 2. Material and Methods

### 2.1. Patients

All participants provided informed consent prior to recruitment. No personal/identifiable information was used.

The Leeds Inflammatory Arthritis Disease Continuum (IACON, National Research Ethics Service, West Yorkshire Ethics Committee REC approval: 09/H1307/98) is a longitudinal cohort register (2010–2015), comprising over 1500 participants with early IA symptoms with <24 months of symptoms. Patients were followed for 2 years until they fulfilled the ACR-87/EULAR-2010 classification criteria for RA. This register continued from 2015 (Rheumatoid Arthritis DiseAse Research (RADAR), REC:09/H1307/98); however, in accordance with progression in clinical practice, only the EULAR-2010 RA criteria were used, and records excluded patients with symptoms that resolve over 1–3 months. Data records in the registers include age, gender, smoking, symptoms duration at inclusion, presence of autoantibodies (RF/ACPA), tender/swollen joints count (TJC/SJC), C-reactive protein level (CRP), and a calculated disease activity score (DAS28-CRP).

### 2.2. Sample Preparation

Whole blood cells, PMBC, and CD4+ T-cells were used. EDTA samples of peripheral blood were collected from participants. PBMC were isolated by lymphoprep density gradient separation according to the manufacturer’s instructions and stored in liquid nitrogen for future use. CD4+ T-cells were isolated from frozen PBMC using immuno-magnetic negative selection (EasySep Human CD4+ T Cell Isolation Kit (STEMCELL Technologies, Cambridge, UK) according to protocol). After isolation, the purity of the isolated CD4+ T-cells was verified by flow cytometry at >95%. The cells were then processed for DNA extraction.

### 2.3. DNA Preparation and Bisulfite Conversion

Genomic DNA was extracted using a silica-membrane-based DNA purification kit (QIAamp DNA Blood Mini Kit (Qiagen, Hilden, Germany), according to the manufacturer’s protocol (detailed in Appendix A). DNA quality and concentration were assessed using a spectrophotometer (ND1000, Thermo Fisher Scientific, Wilmington, NC, USA). To differentiate between methylated and unmethylated cytosine by PCR, bisulfite conversion of the DNA is necessary to differentiate unmethylated cytosine turned into uracil while keeping methylated cytosine untouched. Bisulfite conversion was performed using the EZ DNA-Methylation-Gold Kit (Zymo Research, Irvine, CA, USA), following the manufacturer’s protocol (detailed in Appendix A). The quantity of DNA after bisulfite conversion was again measured by nanodrop.

### 2.4. Primer Design for qMSP

The subsequent design of the qPCR assay relies on the same principles as other fluorescent-probe-based assay using primers to ensure the specificity of the PCR product and a probe to quantify it. The TaqMan^®^ MGB Probe and primer sets were designed using Primer Express Software v2.0. Details of the design are provided in the Appendix A. All primers were synthesised by Thermo Fisher Scientific and reconstituted to a concentration of 100 μM in sterile water. TaqMan custom probe MGBNFQ was synthesised by Applied Biosystems (Waltham, MA, USA) and reconstituted to a concentration of 2.5 μM in sterile water. Primers and probes were aliquoted and stored at −20 °C. The details of each design are described in the Section 3.

### 2.5. qMSP Assay Optimisation

The development of qMSP assays starts with optimisation of the reactions by varying the concentration of primers to ensure both specificity (methylated/unmethylated DNA) and efficiency (equivalent yield for each PCR product). For this, reactions were set up by varying primer concentrations and then directly comparing Cts between assays. A dilution series of the template DNA (0.2 to 50 ng) was then used to compare the efficiencies of the different assays.

### 2.6. Statistics

Continuous variables were not normally distributed. Data are described using the median and interquartile range (IQR). Non-parametric tests were used throughout. To compare differences between groups, the Mann–Whitney U (MWU) and Kruskal-Wallis tests were used for continuous variables and Chi2 for categorical variables. No adjustments for multiple testing were made at the univariate level. Covariance was analysed using the point biserial correlation and Spearman correlation tests. Binary logistic regression was used to determine the relationship of different variables with RA classification. Odd ratio (OR) and receiver operating characteristics (ROC) analyses were performed to establish the predictive value of individual variables and models, along with performance scores for sensitivity and specificity, as well as positive and negative predictive values (PPV/NPV). For multivariate analysis, a binary logistic regression using a forward approach was used to select the variables to build models. The models with and without the TNF-qMSP data were then compared. Bootstrapping was performed to generate a more reliable confidence interval of the model’s area under the ROC curve (AUC). Analyses were conducted using R software (version: x64 4.2.1) and SPSS software V27.0 (IBM, Armonk, NY, USA). The level of significance for *p*-values was set at *p* < 0.05.

## 3. Results

### 3.1. Candidate CpG Selection

Our assay design revolves around two fundamental criteria. First, the specificity for a particular cell type applying criteria needed for a successful qMSP assay based on a specific template [12,40]. Given the pivotal role of CD4+ T-cells in the pathogenesis of RA, we prioritised CpGs that exhibit differential methylation in RA naïve T-cells (i.e., the target cells in the qMSP assay). However, working with clinical samples, including peripheral blood mononuclear cells (PBMCs) and whole blood (WB), necessitates considering a second criterion in the design: the practical utility of the assay in clinical settings. The presence of methylation signals from other cell types (non-target cells) in the samples will potentially interfere with or dilute the primary signal, making it challenging to observe significant differences in methylation of the CpG investigated. We included an illustration of the principles and constraints of designing qMSP assays in the Appendix A. To address this challenge, we employed two strategies for selecting candidate CpGs.

Strategy 1: We made use of our data (GSE121192), with naïve CD4+ T-cells being our target cells and the non-target cells being monocytes as representative of possible contaminating cells in the samples. From CpGs differentially demethylated in naïve CD4+ T-cells (2891 CpG, all *p*-value ≤ 0.001), we selected those with a low methylation β-value comprised between 0% and 50% that were also highly methylated in monocytes (β-value > 80%), resulting in 606 candidates. We then prioritised the CpGs with the highest difference in levels of methylation between health and RA, with a Δβ-value > 5% (*p*-value ≤ 0.0001). This resulted in 26 candidate CpGs (Appendix A), ranked at the top for their relevance to the RA disease process. The top genes were TNF-α, followed by IFMT1 (involved in type-I interferon signalling), both of which have relevance in RA. There was a Δβ = 22% between HC and RA for the TNF gene and Δβ = 20% for the IFMT1 gene (Appendix A).

As the limitation of this first strategy is its high specificity for T-cells, it may not allow for interference from other cells in the samples; hence, it may be necessary to purify the CD4+ T-cells to get robust data. Consequently, we developed a second strategy aiming to identify CpGs displaying substantial differences in methylation levels between HC and RA in the target cells, thus allowing for some interference from other cells in the samples (i.e., PBMC or WB).

Strategy 2: We search publicly available DNA methylation datasets using the 450k DNA methylation arrays for additional cell types, such as CD4+ and CD8+ T-cells, B and NK-cells, monocytes for PBMC, and neutrophils, eosinophils, and granulocytes for WB (detailed in Appendix A) from healthy control (HC) and any datasets in RA patients (early and established disease). We selected CpG candidates from our naïve CD4+ T-cell dataset, filtering CpGs, with a large size effect Δβ-value ≥ 10% between HC and RA patients (*p*-value ≤ 0.001). This resulted in 1508 possible candidates that needed to be refined and prioritised. We then sought CpG that exhibited a high β-value (≥50%) in other cell types (i.e., CD8, NK, and B cells or monocytes) to limit the dilution of the signal from CD4 T-cells. This resulted in 22 CpG candidates (listed in Appendix A), although no perfect candidate could be identified and interference from other cell types appeared to be unavoidable. Similar filtering was attempted using neutrophils, eosinophils, and granulocyte datasets, which eliminated all candidates and therefore was not considered further in the final selection. The gene at the top of the ranking was histone deacetylase 4 (HDAC4, epigenetic histone modifications, with 3 candidate CpGs, Δβ = 12–19%, although with relatively low β-value in most cell types); followed by IRF8 (type I Interferon signalling, 1 CpG, Δβ = 14%, high β-value in most cell types); and then micro-RNA (MIR21, 3 candidate CpGs, Δβ = 11–19%, and one CpG with a higher β-value in most cell types except NK-cells).

These 26 and 22 candidates were then further considered with respect to the local methylation pattern of the region surrounding the selected CpG, which is critical for the design of the qMSP assay. We examined the surrounding 300–500 pb region to allow for other CpGs to contribute to the primer/probe design and enhance the specificity of the assay. Appendix A illustrates methylation profiles in the region of the TNF-α gene with respect to strategy 1 in the top panel and strategy 2 in the bottom panel.

We decided to select 2 genes from the 1st strategy (TNF-α and IFITM1), which would in principle require purified CD4+ T-cells, and 3 from the second strategy (HDAC4, IRF8, and MIR21), allowing for the use of PBMC. This was also taking into consideration the biological relevance of these candidates to RA, as we previously reported on changes in expression of IFITM1 (hypo-methylated and over-expressed) and IRF8 (hyper-methylated and under-expressed) in CD4+ T-cells in early RA compared to HC [39] (Appendix A), while for TNF-α, changes were observed but were more limited (hypo-methylated but non-significantly over-expressed). MIR21 has been reported as well in RA, with >5-fold higher expression levels, but in the serum rather than cells [41].

### 3.2. Primer Design, Optimisation, Specificity of the Assay and qPCR Efficiency

We proceeded to primer design for 4 genes at the top of our selection lists (TNF-α, IRF8, HDAC4, and MIR21) and an internal control gene for normalisation (GAPDH). The IFIMT1 assay was designed but not pursued, as sequencing data (performed as described for the TNF gene [39]) confirmed a trend for overall lower methylation in RA patients, but with very high heterogeneity. Due to the lower complexity of bisulfite-converted DNA, a higher GC content in methylated DNA allows for the high primer’s Tm optimal for qPCR (60 °C). We therefore designed assays to quantify methylated CpG. Primer pairs designed for the assays retained are detailed in Appendix A. The sequence surrounding both of the MIR21 CpG candidates allows for primer design, although this assay could not be optimised satisfactorily as sufficient specificity for methylated DNA could not be achieved.

Primer optimisation was performed to establish conditions for the best amplification using 100% methylated control DNA, choosing conditions for achieving the lowest Ct value for the genes of interest that were also equivalently amplified for the reference gene GAPDH. All combinations of primer concentrations gave very similar results for the TNF candidate CpG (Figure 1A,B), while for the GAPDH gene, the R-primer had a clear limiting effect on the amplification, independent of the amount of F-primer. For the IRF8 gene, both F and R primers had a limiting effect. Further details are provided in the Appendix A. The final conditions used for all assays are listed in Appendix A.

The specificity of the assay for methylated DNA was then verified using 100% unmethylated control DNA. No amplification should be observed for the target genes, in contrast to the assay for the reference gene GAPDH, which needs to be methylation-independent. This was achieved for the TNF and HDAC4 genes but not for IRF8, where some amplification was observed with the unmethylated control DNA (Figure 1C).

The efficiency of the qPCR assays for the target gene and internal control gene needs to be equivalent for a relative quantification method. This was assessed using a wide range of input DNA concentrations (Figure 1D, from 0.2 ng to 50 ng for the GAPDH and TNF assays). The Ct value from each DNA input was plotted against Log (DNA concentration ng/μL). Fitting a standard curve to a linear regression model provided the slope used to calculate qPCR efficiency. The regression showed true linearity for all assays over the full range of concentrations used, demonstrating that the amount of input DNA would have no effect on the results. qPCR efficiencies ranged from 95.7% to 99.9%, suggesting a true 2-fold amplification at each PCR cycle for the TNF, HADAC4, and GAPDH genes. The difference in efficiency between 100% methylated and unmethylated control DNA was also <1% with the GAPDH gene assay.

For the IRF8 assay, the amplification showed linearity on control DNA, although on a reduced range (from 1 to 20 ng of DNA). The efficiency was outside of the acceptable value (124.45%), suggesting >2-fold amplification per cycle. The IRF8 assay was not optimised further. Other sets of primers showed similar issues.

With the technical validation steps being verified for those 3 assays, a relative quantification method could be adopted to compare the methylation of the TNF or the HDAC4 gene. A final control was then added, with 100% methylated control DNA reaction to each PCR plate as a reference to ensure the repeatability of the PCR reactions between plates (further details are provided in the Appendix A).

### 3.3. Template DNA: CD4+ T-cells, PBMC or WB

For a practical test, the assay’s robustness would benefit from the least processed input material (i.e., DNA) in order to limit the variability introduced during multiple processing steps. To establish the most suitable source of DNA, purified CD4+ T-cells, PBMCs, and WB were compared in a small group of RA patients and HCs. The 3 types of samples could not be obtained from the same individual, but donors were carefully age-matched.

We first confirm the highly significant difference in DNA methylation between HC and RA in purified CD4+ T-cells (Figure 1E, data for the TNF assay, HC (n = 5) and RA (n = 6), *p* = 0.002). In PBMC, there was less difference (*p* = 0.015), while in WB, no significant difference could be observed anymore (4 HC and 8 RA samples). The lack of difference in levels of methylation observed in WB is likely to result from a loss of signal due to a dilution effect, with CD4+ T-cells representing only a small fraction of cells in WB while being about half of cells present in PBMC. Results for the HDAC4 gene are presented in Appendix A.

PBMCs, which are the least processed type of sample allowing discrimination, were chosen to further study the value of the TNF-qMSP assay in early IA patients.

### 3.4. Methylation Levels of the TNF Gene in RA Versus other Arthritis

A total of 127 PBMC samples from patients registered into the IACON study were obtained from our tissue bank. After a 2-year follow-up, 64 patients were classified as RA (ACR-197/EULAR 2010 criteria), of which 23 were classified after up to 15 months of delay (median 6.5 months). Sixty-three were non-RA (including 11 reactive arthritis, 36 undifferentiated arthritis (UA), and 16 psoriatic arthritis (PsA)). The 40/63 non-RA patients were classified with a median delay of 14 months (up to 24 months). Table 1 describes the demographic and clinical data in both groups.

DNA was isolated, bisulfite converted, and used in the TNF and GAPDH qMSP assays. Results (plotted as levels of methylation (%)) were significantly different between diagnosis groups (Figure 2A, Kruskal-Wallis test *p* = 3.5 × 10^−9^ followed by Dunn’s multiple comparison test). There was a significant reduction in RA methylation levels compared with HCs (*p* = 1.97 × 10^−5^), reactive arthritis (*p* = 4.21 × 10^−4^), and UA (*p* = 4.63 × 10^−7^), but not with PSA (*p* = 0.150), despite showing lower methylation levels compared to other groups. There was no significant difference between HCs and reactive arthritis, or UA. Methylation levels between RA (in the left of Figure 2B, median 3.42% [95% CI 2.42; 4.92]) versus non-RA patients (6.44% [4.83; 8.39]) were highly significant (*p* = 4.0 × 10^−9^), confirming that the TNF-qMSP assay has the inherent ability to discriminate between patients progressing to RA from other forms of IA. There was no correlation between methylation levels of the TNF gene and any demographic or clinical variables, with the exception of age, which showed a weak trend for reduced methylation with age (rho = −0.440). This was therefore suggesting that the qMSP data may have independent added biomarker value for RA classification. Lower levels of methylation predicted RA with an OR = 0.62 (95% CI 0.508–0.755, *p* = 2.1 × 10^−6^) and a good AUC = 0.198 (0.120–0.275) (in the left of Figure 2C).

Dichotomizing TNF methylation levels for high versus low risk for progression to RA (at 80% specificity) defined a cut-off at 4.5% of DNA methylation for the low-risk group (above) and a high-risk group (below), creating a new categorical variable. The OR for being RA in the high-risk group was 8.4 (3.77–19.0), sensitivity 68.7%, specificity 79.4%, PPV 77.2%, and NPV 71.4%. The AUC = 0.741 (0.664–0.816) was slightly less performant than when using continuous levels of methylation.

A similar analysis using the HDAC4 qMSP assay is described in Appendix A, showing the limited value of this marker due to significant changes but of a limited amplitude (Appendix A).

### 3.5. Validation in a Second Group

We repeated this analysis in a group of 157 patients from the RADAR register (in the right of Figure 2B), with 126 RA and 31 non-RA patients (4 PsA and 27 UA). The 2 RA groups showed no difference in any variables (Appendix A), while the non-RA groups showed differences for demographic and clinical data due to the change in recruitment strategy focusing on RA/UA rather than all forms of IA (importantly excluding non-persistent IA). This resulted in reduced significance levels for most parameters for their association with RA classification in RADAR compared to IACON (Table 1, 100–1000-fold lower *p*-values).

There was no difference in the distribution of methylation levels in RA patients between the 2 cohorts, while a non-significant small reduction was observed between the 2 non-RA groups (*p* = 0.099). Low levels of methylation predicted RA with an OR of 0.49 (0.372–0.654), *p* = 2.09 × 10^−6^, and AUC = 0.166 (0.072–0.259). TNF methylation levels were consistently dichotomized at the same cut-off (4.5% of methylation) for 80% specificity for RA classification. The OR for being RA with a high-risk result was 27.6 (9.46–80.3, *p*-value = 1.2 × 10^−12^), sensitivity/specificity 84.1%/83.8%, PPV/NPV 95.4%/56.5%, and the AUC = 0.840 (0.767–0.913).

Both cohorts therefore demonstrated similar classification performances for the TNF qMSP assay, despite the changes in clinical practice over time. The combined dataset (n = 284, overall descriptive characteristics in Appendix A) showed an OR of 0.54 (0.461–0.642), *p* = 5.6 × 10^−13^, and an AUC of 0.171 (0.115–0.227), in the left of Figure 2C.

Being in the high-risk group for TNF DNA-methylation was frequently observed in RA (150/190 (79%) compared to non-RA (18/94 (19%)), with a high predictive value (OR = 15.8 (8.25–28.25), *p* = 2.2 × 10^−22^, AUC = 0.799 (0.750–0.848)). The TNF-qMSP data, therefore, showed excellent predictive performance and may improve classification models using demographic and clinical parameters.

### 3.6. Added Value of the TNF Gene Methylation for RA Classification

Several demographic and clinical parameters are known to be associated with RA classification [42,43,44,45]. We performed a multivariate analysis on the combined 284 patients for potential confounders for the prediction of RA classification and to establish the possible added value of the TNF qMSP data. DAS28 was calculated and used for modelling (instead of its constituents) to increase power. Unadjusted OR (*p*-value) for all parameters is presented in Table 2, and all variables showed potential value apart from symptom duration, smoking, and gender.

Binary logistic regression was performed using a stepwise forward approach, allowing free selection of the best predictors sequentially (Table 2). A reference model was built including demographic and clinical parameters with 4 incremental steps, including ACPA, age, and DAS28 (all *p* < 0.0001), followed by RF (*p* = 0.0011). This allowed us to predict RA classification with 85.2% accuracy and an AUC of 0.917 (0.882, 0.952). The added value of the TNF-qMSP assay was then tested. The model first selected the TNF-qMSP data (*p* < 0.0001), then ACPA (*p* = 0.0001), age (*p* < 0.0001), DAS28 (*p*=0.0008), and lastly RF again (*p* = 0.0006). This improved the model’s accuracy by +2.5% (87.7%) and the AUC by +2.7% (in the left of Figure 2D, 0.944 (0.919, 0.969)). A model based on TNF risk-group increased accuracy further by +2.8% (88.03%) and AUC by +3.7% (0.954 (0.931, 0.976)) when selecting the same variables.

To further confirm the added value of the TNF-qMSP assay, we validated the significance of the difference between the AUCs of the reference model and the TNF model. Bootstrapping was used as a resampling technique to create 2000 bootstrapped datasets from the combined dataset. It calculated an AUC 2000 times. The distributions of AUCs for both models had means of 0.9183 (0.9175–0.9191) and 0.9470 (0.945–0.947), respectively. The absence of overlap between the confidence intervals of two distributions of AUCs confirmed a significant added value of the TNF model (*p* < 2.2 × 10^−16^ by T-test).

### 3.7. Clinical Value of RA Classification in ACPA Negative Patients (n = 167)

Patients positive for ACPA are very likely to develop RA [10]; however, only 50–60% of people with RA are positive in population-based cohorts [9,10,11]) and in line with our cohort (108/190, 58%). A novel classification biomarker for ACPA-negative patients would be of great clinical importance. The analysis process was therefore repeated using only ACPA-negative patients from the combined datasets (RA n = 81, non-RA n = 86, descriptive characteristics in Appendix A). RF was observed in 34/167 cases (20%) with an OR = 7.04 (2.73, 18.2) but a poor AUC = 0.638 (0.579–0.696). The TNF methylation levels differed between the RA and non-RA groups (*p* = 9.8 × 10^−16^), OR = 0.45 (0.34, 0.58) and AUC = 0.140 (0.079–0.201) (in the right of Figure 2C). Being in the high-risk group for TNF DNA-methylation was frequently observed with a high predictive value with an OR of 24 (10.6–54.6), *p* of 2.2 × 10^−22^, sensitivity of 86.4%, specificity of 79.1%, PPV/NPV of 79.5/8.1%, and AUC of 0.827 (0.770–0.885).

Unadjusted OR was calculated (Table 3), and variables that predicted RA were age, RF, DAS28, and TNF-qMSP data, while gender, symptom duration, and smoking were not predictive in this group.

Models were developed using a similar approach (Table 3). The reference model predicted 79.6% of cases with an AUC of 0.892 (0.845, 0.939). Adding the TNF DNA methylation levels increased the accuracy by +3% (82.6%) and the AUC by +8.5% (in the right of Figure 2D, AUC = 0.9299 (0.895, 0.965). TNF risk groups also increased prediction by 8.6% for accuracy (88.2%) and by +5.3% for AUC (AUC = 0.954) (0.914, 0.976).

Bootstrapping was performed, and the difference between the mean AUCs for the reference model 0.8929 (0.8919–0.8940) and that of the TNF model 0.9337 (0.9329–0.9345) was significant (*p* < 2.2 × 10^−16^).

## 4. Discussion

We selected several candidate CpGs and fully developed qMSP assays from genes with a probable biological role in RA. One assay proved to be limited by the gene sequence (MIR21), another by inadequate assay performance (IRF8), while satisfactory technical development was achieved for 3 candidates (TNF, HDAC4, and GAPDH), although the validation in clinical samples showed limited value for 1 gene (HDAC4). The TNF-qMSP data showed high individual performance for the classification of RA patients, and when combined with clinical variables, it improved the classification performance in both the overall cohort and in ACPA-negative patients. This demonstrated that the TNF-qMSP assay has added value and the potential to be used as a diagnostic biomarker.

We chose the qMSP technology for an epigenetic biomarker measurement assay because it is simple, permits the use of blood, and imposes a number of conditions on the suitability of the candidate CpG. Designing a robust qMSP assay can therefore be challenging, as further discussed more comprehensively in Appendix A. qMSP design principles have been successfully employed here to develop the TNF-qMSp assay in PBMCs (but not WB), while our data for the other genes illustrate the challenges encountered. qMSP design can nonetheless be successful using WB, for example, quantifying regulatory T-cells (using the Foxp3 gene) and Th17 cells (using the IL17A gene) [46], thanks to the unique cell-type specificity of the region targeted in both those assays. Once well developed, qMSP is highly sensitive, efficient, and reliable compared to many other DNA-methylation analyses. It is also cost- and time-effective, which is important for high-throughput screening in biomarker research. The selection of the TNF gene as the best candidate for the design of a qMSP assay was imposed by the constraints of the assay design, as well as its high relevance to RA pathology. We previously demonstrated that the promoter of this gene is differentially methylated in RA (using both 450K array and direct bisulfite sequencing, as shown in Appendix A) early in the disease process [39] and used public transcriptomic data in total CD4+ T-cells (between HC and RA) and immunoassays (performed in house on serum samples) to show that de-methylating the DNA in early RA could be indirectly associated with higher expression (mRNA levels) and higher protein released in the serum (ELISA) [39]. However, to prove causality between changes in DNA methylation and changes at the transcriptional level, transcriptomic data using blood-derived cells are not the best as cells are in a resting state [39]. Cells (notably immune cells) need to be “activated” with appropriate signals to initiate the transcription of genes. As such, experiments using blood CD4+ T-cells showed no (low levels of) detectable basal expression of cytokines by intracellular flow cytometry (IL2, IL10, IFN, TNF, or IL17) between HC and RA, while following activation, a higher percentage of positive cells were indeed observed in RA [46,47,48]. Polarising cells (towards Th17, for example) would further increase the frequency of committed cells while not affecting the levels of expression in individual cells [46]. qMSP assays will therefore allow monitoring of the time when epigenetic commitment events occur in cells, with important utility for identifying the timing of such events over the pre-clinical phases of RA.

Modelling for RA classification showed added value for the TNF-qMSP data. We used DAS instead of its components, which allowed for a better prediction model with this number of cases. For both the overall cohort and the ACPA-negative patient group, the TNF-qMSP data alone was already the best individual predictor of RA classification. Improvement in multivariate models, including TNF-qMSP data, may appear small (+2.7% AUC); however, it results in a +2.5% of patients accurately classified. The gain with the TNF high-risk dichotomization data was larger (+2.8% accuracy and +3.7% AUC), with 151/170 (89%) cases with a high-risk actually progressing to RA compared to 39/114 (34%) in the low-risk group. The gain in NPV (+3%) was also important and would allow non-RA cases to be more rapidly classified, discharged, or directed to the appropriate clinic/management. The +3% gain in PPV overall and +7% in the ACPA-negative group would allow patients with RA to be classified and access treatment at inclusion rather than later (median 6 months of delay), notably for ACPA-negative patients. This demonstrates that TNF-qMSP data have added value and the potential to be used as a diagnostic biomarker. The limitations of the statistical analysis in our work are related to the cohorts studied. Despite being recruited using ACR 1987 and then EULAR 2010 criteria, we were able to show that the biomarker behaves the same in RA patients from both cohorts (showing no shift in b-values), while for the non-RA group in RADAR (27/31 UA), the data were indeed aligned with the UA subgroup in IACON. This allowed us to combine groups in the validation statistical approach and to use bootstrapping to confirm the added value of the qMSP data. We chose a forward regression approach as the 4 steps in the reference model clearly indicated the gain obtained from using ACPA alone to ACPA+1, 2, and 3 variables for predicting RA, while the change incremented by adding the qMSP data to the variable list was that qMSP was selected as the 1st step and before ACPA, then adding other variables. A backward regression approach eliminating variables not contributing to the model resulted in the same variables being selected with limited differences in accuracy, OR, and AUC in the final models (Appendix A). External validation will remain necessary.

## 5. Conclusions

Overall, our data confirmed that the change in DNA methylation is an important feature of cellular events at the molecular level in RA, with high cell-type specificity. Designing qMSP assays is not trivial, but our TNF-qMSP assay provides a novel tool for RA classification, notably for ACPA-negative patients, which represent most of the population of RA patients experiencing delays in diagnostics.

## 6. Key Messages

RA classification is an important field for biomarker research in early disease, and while many biomarkers are proposed, few achieve clinical utility.We demonstrate that a biomarker selection approach based on biological rationale and solid observations, added to publicly available knowledge, allows for successful and robust designs.Overall, our study provides a novel tool for RA classification, notably for ACPA-negative patients, which can reduce the time to diagnosis and enable earlier access to treatment.

## Figures and Tables

**Figure 1 cells-12-02376-f001:**
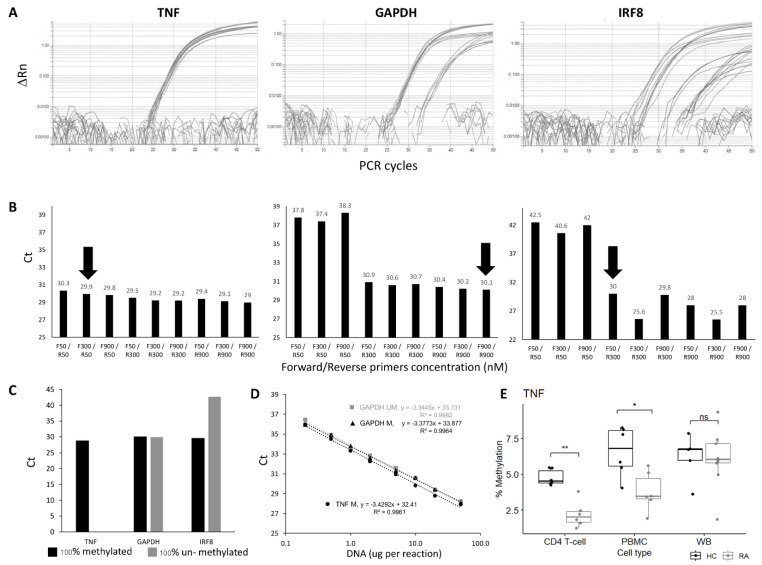
qMSP assay optimisation and verification. (**A**) Amplification plot and (**B**) CT values of TNF, GAPDH, and IRF8 qMSP assays using forward and reverse primers at different concentrations (50, 300 or 900 nM). The primer concentrations at F300/R50 and F900/R900 and F50/R300 nM were chosen for TNF, GAPDH, and IRF8 assays, respectively (black arrow). (**C**) CT values of TNF, GAPDH, and IRF8 assays using 100% methylated control DNA and 100% unmethylated control DNA as templates. (**D**) Dilution curve and best-fit line of 2 qMSP; GAPDH assay, for methylated and unmethylated control DNA and the TNF assay, the 100% methylated control DNA only showing amplification. Both assays show good efficiency and a good fit of the regression model to the data. (**E**) Boxplot of the % of methylation of the *TNF* qMSP in CD4+ T-cells, PBMC, or WB in early RA (Grey, n = 6) and healthy control (black, n = 6). MWU test: ** *p* < 0.001, * *p* < 0.05, ns: not significant.

**Figure 2 cells-12-02376-f002:**
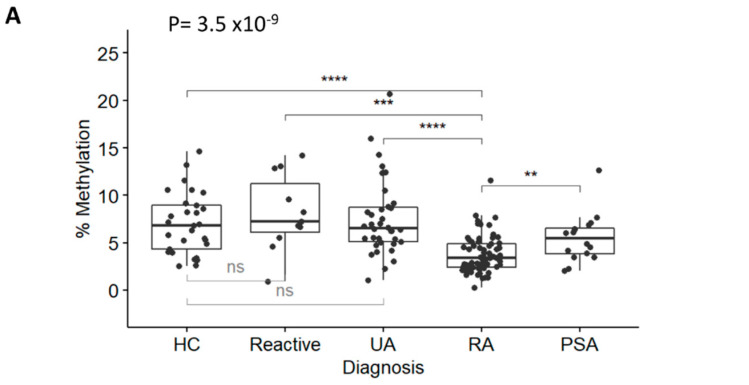
RA classification using the TNF-qMSP assay. (**A**) Box plot representation of the levels of DNA-methylation (%) for the TNF assay in IACON samples (n = 127) for 4 diagnostic groups and HCs. Kruskal-Wallis test *p*-values, followed by Dunn’s multiple comparison test: **** *p* < 0.0001, *** *p* < 0.0001, ** *p* < 0.001 were shown. (**B**) Box plot representation of the levels of DNA methylation (%) in IACON (left panel, RA n = 64 versus non-RA n = 63, MWU *p* = 4.0 × 10^−9^), and RADAR (right panel, RA n = 126 versus non-RA n = 31, *p* = 8.2 × 10^−9^). (**C**) Predictive value (AUROC analysis) of the TNF DNA-methylation levels (black line) and TNF DNA-methylation risk category (grey line) in cohorts combined (left panel, n = 284) and in the ACPA-negative patients (right panel, n = 167). (**D**) Predictive value (AUROC analysis) of the clinical model without (grey line) and with the TNF qMSP data (black line) or with the TNF-qMSP risk category (dotted line) in the combined dataset (n = 284) and in the ACPA-negative patients (n = 167). AUC values are indicated on the graph.

**Table 1 cells-12-02376-t001:** Patient characteristics.

IACON	Missing Data (n)	RA (n = 64)	Non-RA (n = 63)	*p*-Value	AUC (95% CI)
TNF methylation levels *	0	3.42 [2.42; 4.92]	6.44 [4.83; 8.39]	4.0 × 10^−9^	0.198 (0.120; 0.275)
TNF methylation high-risk group, n (%)	0	44 (68.8%)	13 (20.6%)	1.4 × 10^−7^	0.741 (0.664; 0.816)
Age * (years)	0	61.5 [46.8; 73.0]	40.0 [33.5; 49.0]	1.9 × 10^−9^	0.804 (0.728–0.881)
Female gender n (%)	0	51 (79.7%)	45 (71.4%)	0.385	0.540 (0.439; 0.641)
Symptoms duration (months) *	2	5.0 [2.0; 10.0]	6.0 [3.0; 12.0]	0.253	0.442 (0.341; 0.544)
Smokers n (%)	10	40 (66.7%)	25 (43.9%)	0.0018	0.606 (0.516; 0.696)
RF positive (%)	3	31 (48.4%)	6 (9.84%)	5.9 × 10^−6^	0.690 (0.617; 0.764)
ACPA positive (%)	4	29 (48.3%)	2 (3.17%)	2.7 × 10^−8^	0.723 (0.655; 0.792)
Tender joint count *	0	9.00 [5.00; 15.2]	2.00 [1.00; 6.00]	1.5 × 10^−8^	0.774 (0.686; 0.861)
Swollen joint count *	0	5.50 [1.75; 9.00]	1.00 [0.00; 2.00]	3.6 × 10^−8^	0.763 (0.676; 0.849)
CRP * (mg/L)	10	12.0 [<5; 23.2]	<5 [<5; 10.8]	9.0 × 10^−4^	0.667 (0.574; 0.760)
DAS28 *	10	5.05 [3.68; 5.62]	2.90 [2.30; 3.95]	1.1 × 10^−9^	0.801 (0.715; 0.887)
RADAR	Missing data (n)	RA (n = 126)	Non-RA (n = 31)	*p*-value	AUROC (95% CI)
TNF methylation levels *	0	3.24 [2.52; 4.10]	5.15 [4.81; 6.89]	8.2 × 10^−9^	0.166 [0.072; 0.259]
TNF methylation high-risk group, n (%)	0	106 (84.1%)	5 (16.1%)	4.8 × 10^−13^	0.840 [0.767; 0.913]
Age * (years)	0	57.5 [49.2; 68.0]	52.0 [35.5; 61.5]	0.036	0.608 (0.484; 0.729)
Female fender n (%)	0	86 (68.3%)	20 (64.5%)	0.854	0.522 (0.401; 0.643)
Symptoms duration * (months)	6	6.42 [3.67; 11.9]	5.96 [3.67; 8.49]	0.468	0.560 (0.450; 0.670)
Smokers n (%)	2	77 (61.1%)	17 (54.8%)	0.664	0.520 (0.406; 0.634)
RF positive n (%)	2	70 (55.6%)	4 (12.9%)	4.9 × 10^−5^	0.710 (0.617; 0.802)
ACPA positive n (%)	2	80 (63.5%)	6 (19.4%)	2.4 × 10^−5^	0.718 (0.621; 0.814)
TJC *	0	10.0 [3.00; 15.8]	4.50 [2.00; 10.0]	0.083	0.610 (0.495; 0.726)
SJC *	0	4.00 [2.00; 9.00]	3.00 [1.00; 4.00]	0.070	0.611 (0.508; 0.715)
CRP * (mg/L)	10	11.2 [<5; 23.0]	5.00 [<5; 12.3]	0.027	0.614 (0.503; 0.725)
DAS28 *	10	4.43 [3.45; 5.31]	3.73 [2.99; 4.30]	0.006	0.666 (0.566; 0.766)

Data are described using n (% of patients) or * median [Inter Quartile Range limits]. RF rheumatoid factor, ACPA anti-citrullinated peptide antibodies, TJC tender joints count, SJC swollen joints count, CRP C-reactive protein (lower limit of detection is <5 mg/L), DAS28 disease activity score 28 joints.

**Table 2 cells-12-02376-t002:** Binary logistic regression (n = 284) forward method.

OR (95% CI) *p* Value	Unadjusted	Reference Model	TNF qMSP Model (Levels)	TNF qMSP Model (Risk Groups)
TNF methylation levels	0.54 (0.46, 0.64) <0.0001		0.54 (0.42, 0.68) <0.0001	
TNF methylation high-risk group	15.8 (8.51, 29.5) <0.0001			19.05 (7.87, 51.41) <0.0001
Age	1.06 (1.04, 1.08) <0.0001	1.07 (1.05, 1.10) <0.0001	1.06 (1.03, 1.09) 0.0001	1.06 (1.03, 1.09) 0.0003
Female gender	1.15 (0.67, 1.98) 0.6052	not selected	not selected	not selected
Symptoms duration	0.99 (0.97, 1.01) 0.278	not selected	not selected	not selected
Smokers	2.17 (1.31, 3.59) 0.0255	not selected	not selected	not selected
RF positive	9.53 (4.66, 19.5) <0.0001	4.42 (1.83, 11.31) 0.0011	3.96 (1.53, 11.09) 0.0006	4.97 (1.77, 11.54) 0.0033
ACPA positive	14.5 (6.64, 31.5) <0.0001	18.92 (7.41, 54.99) <0.0001	26.35 (8.97, 92.87) <0.0001	33.38 (10.53, 129.28) <0.0001
DAS28	2.01 (1.62, 2.51) <0.0001	1.84 (1.41, 2.45) <0.0001	1.66 (1.25, 2.25) 0.0008	1.78 (1.32, 2.47) 0.0003
Accuracy (% of correctly predicted cases)		85.21 (80.54, 89.13)	87.67 (83.28, 91.26)	88.03 (83.67, 91.56)
SEN % (95% CI) SPE % (95% CI)		91.05 (86.06, 94.70) 73.40 (63.29, 81.99)	96.16 (88.58, 96.31) 76.59 (66.73, 84.71)	92.10 (87.31, 95.51) 79.79 (70.25, 87.37)
PPV % (95% CI) NPV % (95% CI)		87.37 (81.92, 91.66) 80.23 (70.25, 88.04)	88.94 (83.74, 92.94) 84.71 (75.27, 91.60)	90.21 (85.13, 93.99) 83.33 (74.00, 90.36)
AUC (95% CI)		0.917 (0.882, 0.952)	0.944 (0.919, 0.969)	0.954 (0.931, 0.976)

RF rheumatoid factor, ACPA anti-citrullinated peptide antibodies, DAS28 disease activity score 28 joints. SEN/SPE sensitivity/specificity, PPV/NPV positive and negative predictive value, AUC area under the ROC curve, CI confidence interval.

**Table 3 cells-12-02376-t003:** Binary logistic regression in ACPA negative patients (n = 167).

OR (95% CI) *p* Value	Unadjusted	Reference Model	TNF qMSP Model (Levels)	TNF qMSP Model (Risk Groups)
TNF methylation levels	0.45 (0.34, 0.58) <0.0001		0.50 (0.37; 0.65) <0.0001	
TNF methylation high-risk group	24.0 (10.6, 54.6) <0.0001			28.9 (9.3; 85.1) <0.0001
Age	1.10 (1.06, 1.13) <0.0001	1.09 (1.06, 1.13) <0.0001	1.09 (1.05, 1.13) <0.0001	1.09 (1.05, 1.14) <0.0001
Female gender	1.63 (0.81, 3.30) 0.1781	not selected	not selected	not selected
Symptoms duration	0.99 (0.96, 1.01) 0.336	not selected	not selected	not selected
Smokers	1.84 (0.99, 3.40) 0.054	not selected	not selected	not selected
RF positive	7.04 (2.73, 18.2) <0.0001	8.64 (2.88, 30.54) 0.0002	7.38 (2.17, 30.21) 0.002	9.41 (2.23, 39.68) 0.002
DAS28	2.06 (1.57, 2.69) <0.0001	1.76 (1.31, 2.46) 0.0004	1.56 (1.12, 2.22) 0.0099	1.64 (1.16, 2.40) 0.0072
Accuracy (% of correctly predicted cases)		79.54 (72.73, 85.47)	82.63 (76.02, 88.05)	88.02 (82.11, 92.53)
SEN % (95%CI) SPE % (95%CI)		77.78 (67.17, 86.27) 81.40 (71.55, 88.98)	85.19 (75.55, 92.10) 80.23 (70.25, 88.04)	88.89 (79.95, 94.79) 87.21 (78.26, 93.44)
PPV% (95%CI) NPV% (95%CI)		79.74 (69.20, 87.96) 79.54 (69.61, 87.40)	80.23 (70.25, 88.04) 85.19 (75.55, 92.10)	86.75 (77.52, 93.19) 89.29 (80.63, 94.98)
AUC (95% CI)		0.892 (0.845, 0.939)	0.930 (0.895, 0.965)	0.954 (0.914, 0.976)

RF rheumatoid factor, ACPA anti-citrullinated peptide antibodies, DAS28 disease activity score 28 joints. SEN/SPE sensitivity/specificity, PPV/NPV positive and negative predictive value, AUC area under the ROC curve, CI confidence interval.

## Data Availability

The datasets acquired during this study are available from the corresponding author on reasonable request.

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
