# Peer review of "DNA Hypomethylation in the TNF-Alpha Gene Predicts Rheumatoid Arthritis Classification in Patients with Early Inflammatory Symptoms"

_cells, 2023, doi:10.3390/cells12192376_

Round 1

Reviewer 1 Report

The study describes a new methodology to identify potential biomarkers to predict RA development in patients with unclassified joint complains. The focus here is on methylation of the TNF gene. The same group has published on this qMSP method since 2014 (Ponchel F, ARD 2014).

Table 1 is missing.

The step between the whole genome RNA-seq to selection of the TNF, IRF8; HDAC4, MIR21 and IFIMT1 genes for the CpG methylation analysis is insufficiently described. Thus, the choice is difficult to grasp. How were the available RNA-seq and methylation data sets used?

What is a relation between the methylation of TNF gene, traditional mRNA levels and the protein TNF levels in serum or in PBMC cultures? This important information should be added to the results and the point should be discussed. 

What is "high risk" samples/patients?

Step forward selection strategy with a free choice of potential predictors is used in the study, which could introduce a selection bias. Suggest doing traditional backward stepwise elimination regression instead.

Author Response

Dear Reviewer,

We thank you for your valuable feedback and constructive comments on our manuscript. We have carefully considered your suggestions and have made the necessary revisions to enhance the discussion section of our paper. Below, we address each of your points:

The study describes a new methodology to identify potential biomarkers to predict RA development in patients with unclassified joint complains. The focus here is on methylation of the TNF gene. The same group has published on this qMSP method since 2014 (Ponchel F, ARD 2014).

We apologies for some confusion if we led this reviewer to think we published before on qMSP assay development. We published on several types of  biomarker technologies but the qMSP was developed for the first time in this paper. The paper cited was using flow cytometry.

  • Table 1 is missing.

We deeply apologize for not spotting that Table 1 was not included in the pre-layout print for review. We corrected this.

1.2 The step between the whole genome RNA-seq to selection of the TNF, IRF8; HDAC4, MIR21 and IFIMT1 genes for the CpG methylation analysis is insufficiently described. Thus, the choice is difficult to grasp. How were the available RNA-seq and methylation data sets used?

We did not use RNAseq data for this work. We only used data from genome-wide methylation data (450K Illumina DNA methylation microarray) both from our lab (as published ref 39) and from publicly available sources (line 134). The analysis of differentially methylated CpG candidates between HC and RA has been published (ref 39), generating a list of candidate CpGs. The main criteria allowing the design of a qMSP assay is whether or not candidate CpG are fully-methylated in Non-CD4+T-cells is as it directly impact on the sensitivity of the assays and the material that can be used. The rational for choosing those was that they were best at fulfilling the 2 selection strategies as highly methylated in other cells and differentially methylated between HC and RA in CD4+T-cells. This is described in details line 138-167 of the manuscript and illustrated in the figure for the TNF gene promoter in supplementary material. We designed an additional figure in Sup material to illustrate further the concept and constrains of designing a qMSP assay.

1.3 What is a relation between the methylation of TNF gene, traditional mRNA levels and the protein TNF levels in serum or in PBMC cultures? This important information should be added to the results and the point should be discussed. 

This has been described and discussed in our published manuscript (ref 39). We added more details in the discussion line 395-402.

1.4 What is "high risk" samples/patients?

This is referring to dichotomisation of the TNF methylation levels into 2 categorical groups for high versus low risk as described line 267-270. We added the word “group” to the description and to the table legend to clarify that this is a categorical variable as opposed to continuous TNF methylation levels.

1.5 Step forward selection strategy with a free choice of potential predictors is used in the study, which could introduce a selection bias. Suggest doing traditional backward stepwise elimination regression instead.

We processed a backward regression and obtained the same results with the same 4 markers contributing to the model. We prefer to use a forward selection as the 1st step clearly indicated the gain obtained from ACPA alone to ACPA+1,2,3 variables (forward steps) for the reference model while the change incremented by adding the qMSP to the variable list, was that qMSP was selected 1st and before ACPA, and then +2 other variables. Details were added to the discussion (line 419-425) and data supplied in Table 7 Supplementary file for the backward regression.

Reviewer 2 Report

The article is interesting and very detailed, but some points are omitted and the discussion can be improved. Thus, the study needs a major revision.

- Materials and methods:

- Item 2.1: The authors state that they have C-reactive protein data, however these data are not used in the statistical analyses.

- Item 2.2: Authors must include the full name of all abbreviations used in the article the first time they are mentioned. For example: what is PBMC?

- Item 2.4: This reviewer detected some typing errors, such as: TagMan (line 106). However, other errors may be present, and therefore the authors should revise the entire article (Materiel, ANNOVA....)

- Results:

- Item 3.1: It is imperative that the authors indicate the CpG sites studied in the TNF-alpha: are they in the promoter or exon? What is the position relative to the transcription start site? This information should be used later in the discussion of the results.

- Item 3.4: Table 1 is missing and it is essential for the reader to know the population studied. Authors must include demographic and clinical data from the two cohorts, indicating whether there are statistical differences between demographic and clinical parameters between cohorts and groups.

- Figure 2A: in the legend of the figure, the authors mention thick and thin lines, but in the figure it is not possible to identify differences in the thickness of the lines. It is unclear why the authors used ANOVA followed by Dunn's post test. Anova is a parametric test and Dunn post test is used for non-parametric data (derived from Kruskall-Wallis). Is the data parametric or non-parametric? The data are presented in the text and in the graph as median, so would it be non-parametric data? Please clarify.

Discussion:

- The discussion needs to be improved, for example, the authors do not discuss whether hypomethylation of the studied CpG sites has already been reported in the literature and associated with the level of TNF-alfpa expression. In addition, what is the implication of the level of TNf-alpha expression (higher or lower) with the clinical aspects of RA. It is imperative that the authors indicate the CpG sites studied: are they in the promoter or exon? What is the position relative to the transcription start site?

- The authors also need to mention whether there are already studies evaluating the methylation profile in RA and comparing the results. TNF-alpha hypomethylation has also been reported in other autoimmune diseases?

- The lack of expression analysis is also a limitation of the study.

 Minor editing of English language required.

Author Response

Dear Reviewer,

We thank you for your valuable feedback and constructive comments on our manuscript. We have carefully considered your suggestions and have made the necessary revisions to enhance the discussion section of our paper. Below, we address each of your points:

 - Materials and methods:

2.1- Item 2.1: The authors state that they have C-reactive protein data, however these data are not used in the statistical analyses.

We used DAS28 as a single variable combining its 4 components (TJC, SJC, CRP and GH-VAS) to gain power as cited line 319.  Similar modelling using 4 variables instead of DAS28 showed less predictive value as number of cases was limiting. DAS28 is widely used in clinical practice.

2.2- Item 2.2: Authors must include the full name of all abbreviations used in the article the first time they are mentioned. For example: what is PBMC?

Thank for spotting this omission. We added an abbreviations list.

2.3- Item 2.4: This reviewer detected some typing errors, such as: TagMan (line 106). However, other errors may be present, and therefore the authors should revise the entire article (Materiel, ANNOVA....)

We went through the overall text and hope to have identified any other possible typos (using UK English).

- Results:

2.4- Item 3.1: It is imperative that the authors indicate the CpG sites studied in the TNF-alpha: are they in the promoter or exon? What is the position relative to the transcription start site? This information should be used later in the discussion of the results.

We agree the details are important and they were presented in our previous publication (ref 39). The candidate CpG, namely cg11484872 (located on chromosome 6 (GRCh37/hg19) at the coordinates chr6: 31543169 as shown in SUP-Figure 1). We added more legend to this figure to cover this information. cg11484872was identified in the 450K Methylation Array dataset and is part of the promoter region approximately 200 base pairs upstream of the transcription start site. The primers and probe designed for the TNFα qMSP assay target the region where the cg11484872 is and the PCR product include another 4 CpG sites also similarly differentially methylated as previously described (ref 39).   

2.5- Item 3.4: Table 1 is missing and it is essential for the reader to know the population studied. Authors must include demographic and clinical data from the two cohorts, indicating whether there are statistical differences between demographic and clinical parameters between cohorts and groups.

We deeply apologize for not spotting that Table 1 was not included in the pre-print for review. We corrected this.  There was difference between the 2 groups as stated line 293-300 due to change in inclusion criteria in the register notably for the non-RA groups, associated with the exclusion of patients with non-persistent IA. We added Sup-Table 6 to show that in supplementary file.

2.6- Figure 2A: in the legend of the figure, the authors mention thick and thin lines, but in the figure it is not possible to identify differences in the thickness of the lines. It is unclear why the authors used ANOVA followed by Dunn's post test. Anova is a parametric test and Dunn post test is used for non-parametric data (derived from Kruskall-Wallis). Is the data parametric or non-parametric? The data are presented in the text and in the graph as median, so would it be non-parametric data? Please clarify.

Sorry for incorrectly stating the name of the test. We used the MAN Witney and Kruskal-Wallis test followed by Dunn’s multiple comparison test to compare the different between each group. We have corrected this is the text and legend.

Discussion:

2.7- The discussion needs to be improved, for example, the authors do not discuss whether hypomethylation of the studied CpG sites has already been reported in the literature and associated with the level of TNF-alpha expression. In addition, what is the implication of the level of TNF-alpha expression (higher or lower) with the clinical aspects of RA. It is imperative that the authors indicate the CpG sites studied: are they in the promoter or exon? What is the position relative to the transcription start site?

2.8- The authors also need to mention whether there are already studies evaluating the methylation profile in RA and comparing the results. TNF-alpha hypomethylation has also been reported in other autoimmune diseases?

2.9- The lack of expression analysis is also a limitation of the study.

We agree all points are important details however they were published and discussed in our previous paper, including CpG details, gene region, and sequencing data, RNA levels, and Protein validation (ref 39, some in supplementary material). The biological values of our findings were therefore presented and discussed extensively in this previous publication which was more directly related to pathogenesis.  

Our purpose here is to provide evidence of added value of the qMSP assay as biomarker with clinical utility and discuss the concepts and limitations of the assay design.

We added discussion for these points line 395-402.

Round 2

Reviewer 1 Report

I am not satisfied with the answers regarding the gene selection using genome-wide methylation data. This needs improvement. Were both methods applied to the same material or were the methylation results extrapolated?

I would like to see transcription of the selected/filtered genes in relation to qMSP results.

Author Response

I am not satisfied with the answers regarding the gene selection using genome-wide methylation data. This needs improvement. Were both methods applied to the same material or were the methylation results extrapolated?

We added more details to this section and referred to data presented in SUP Table to illustrate the selection process.

3.1. Candidate CpG Selection

Our assay design revolves around two fundamental criteria. First, the specificity for a particular cell type applying criteria needed for a successful qMSP assay based on a specific template (12, 40). Given the pivotal role of CD4+ T-cells in the pathogenesis of RA, we prioritized CpGs that exhibit differential methylation in RA naïve T-cells (i.e., the target cells in the qMSP assay). However, working with clinical samples, including peripheral blood mononuclear cells (PBMCs) and whole blood (WB), necessitates to consider a second criterion in the design, the practical utility of the assay in clinical set-up. The presence of methylation signals from other cell types (non-target cells) in the samples will potentially interfere with or dilute the primary signal, making it challenging to observe significant differences in methylation of the CpG investigated. We included a figure illustrating the principles and constrains of designing qMSP assays in the supplementary file. To address this challenge, we employed two strategies for selecting candidate CpGs. 

Strategy 1: We made use of our data (GSE121192), naïve CD4+ T-cells being our target cells and the non-target cells being monocytes as representative of possible contaminating cells in the samples. From CpGs differentially demethylated in naïve CD4+T-cells (2891 CpG, all p-value ≤0.001), we selected those with a low methylation β-value comprised between 0% and 50%, that were also highly methylated in monocytes (β-value >80%), resulting in 606 candidates. We then prioritised the CpGs with the highest difference in levels of methylation between health and RA suing a Δβ-value >5% (p-value≤0.0001). This resulted in 26 candidate CpGs (SUP Table 2), ranked at the top for their relevance to the RA disease process. The top genes were TNF-, followed by IFMT1 (involved in type-I interferon signalling), both which have relevance in RA. There was a Δβ = 22% between HC and RA for the TNF gene and Δβ = 20% for the IFMT1 gene (Sup-Table2).

As the limitation of this first strategy is its high specificity for T-cells, it may not allow for interference from other cells in the samples, hence it may be necessary to purify the CD4+T-cells to get robust data. Consequently, we developed a second strategy aiming to identify CpGs displaying substantial differences in methylation levels between HC and RA in the target cell, thus allowing for some interferences from other cells in the samples (i.e., PBMC or WB).

Strategy 2: We search publicly available DNA methylation datasets using the 450k DNA methylation arrays, for additional cell types, such as CD4+ and CD8+ T-cells, B and NK-cells, monocytes for PBMC, and neutrophils, eosinophils, and granulocytes for WB (detailed in SUP Table 1) from healthy control (HC) and any datasets in RA patients (early and established disease). We selected CpG candidates from our naïve CD4+T-cell dataset, filtering CpGs, with a large size effect Δβ-value ≥10% between HC and RA patients (p-value ≤0.001). This resulted in 1508 possible candidates that needed to be refined and prioritised. We then sought CpG that exhibited high β-value (≥50%) in other cell types (i.e., CD8, NK and B cells or monocytes), to limit the dilution of the signal from CD4 T-cells. This resulted in 22 CpG candidates (listed in SUP Table 2), although no perfect candidate could be identified and interferences from other cell-types appeared to be unavoidable. Similar filtering was attempted using neutrophils, eosinophils and granulocytes datasets, which eliminated all candidates and therefore was not considered further in the final selection. The gene at the top of the ranking was histone deacetylase 4 (HDAC4, epigenetic histone modifications, with 3 candidate CpGs, Δβ=12-19%, although with relatively low β-value in most cell types); followed by IRF8 (type I Interferon signalling, 1 CpG, Δβ=14%, high β-value in most cell types); and a then micro-RNA (MIR21, 3 candidate CpGs, Δβ=11-19%, and one CpG with a higher β-value in most cell types, except NK-cells).

These 26 and 22 candidates were then further considered with respect to the local methylation pattern of the region surrounding the selected CpG, which is critical for the design of qMSP assay. We examined the surrounding 300-500 pb region to allow for other CpGs to contribute to the primer/probe design and enhance the specificity of the assay. SUP figure 1 illustrates methylation profiles in the region of the TNF- gene with respect to strategy 1 in the top panel and strategy 2 in the bottom panel.

We decided to select 2 genes from the 1st strategy (TNF and IFITM1), which would in principle require purified CD4+T-cells and 3 from the second strategy (HDAC4, IRF8, and MIR21) allowing for the use of PBMC. This was also taking in consideration the biological relevance of these candidates to RA as we previously reported on changes in expression of IFITM1 (hypo-methylated and over-expressed) and IRF8 (hyper-methylated and under-expressed) in CD4+T-cells in early RA compared to HC ((39) supplementary files), while for TNF, changes were observed but were more limited (hypo-methylated but non-significantly over-expressed). MIR21 has been reported as well in RA with >5 fold higher expression levels but in the serum rather than cells (new REF).

We also compared our dataset in naïve CD4+T-cells with data from naïve CD4+T-cells (personal data communication, although not early, drug naïve RA), total CD4+T-cells (GSE71841) and PBMC (GSE 111942) comparing healthy and RA to select possible CpGs a Δβ-value ≥5%, p<0.05. This resulted in 204 CpGs fitting qMSP design rules in naïve CD4+T-cells, 549 in total CD4+T-cells and 824 in PBMCs. No CpG was common to all 4 datasets and only one (IRF8 gene) was common to 3 lists (ours, total CD4 and PBMC), which was disappointing but illustrate further the challenges of designing qMSP.

This was not detailed in the text as not essential information considering the word count.

I would like to see transcription of the selected/filtered genes in relation to qMSP results

We apologise for not understanding what the reviewer wanted to discuss in the first revision. 

There is limited direct relationships between levels of methylation at individual CpG and gene expression mRNA levels. Hypo versus hyper methylation does not equate to more or less transcription (as discussed in ref 39), but rather reflect the transcriptional capacity of a cell for a certain gene. Altogether, this often results in discrepancies between DNA methylation and gene expression datasets as discussed in our previous paper (39) where we described overlap between DM and DEG (sup data ArrayExpress E-GEOD-20098, E-GEOD-26163, fold change expression ≥1.5, with adjusted p-value ≤ 0.05) with associations between changes in methylation and changes in expression of many genes from the overall TNF pathway rather than the expression of the TNF gene itself in blood CD4+T-cells. IRF8 and IFITM1 were both DM and DEG (ref 39) as well as the TNF gene but not significantly (1.3 fold higher).

However, DNA-methylation changes, particularly in circulating naïve T-cells (i.e. not yet activated) are not sufficient to induce gene expression (particularly for cytokines) but by modifying the accessibility of the DNA to the transcriptional machinery/transcription factors, they can change the capacity or the commitment for expression. Transcription itself would require the addition of transcriptional triggers to initiate mRNA synthesis. As an example, IL2, INF and TNF, IL4 and IL13, or IL17 are not expressed in resting Th1, Th2, Th17-cells respectively (for example <0.1% IL17+ cells measured by flow REF 45). They become detectable in activated cells after a few hours (3-5 fold increase in IL17+ cells, ref 45) reflecting initiation of transcription in more cells but not changes in overall levels of expression in individual cells. The frequencies of IL17+ cells can be further increased when differentiation conditions are used to develop de nono Th17-cells (up to 10 fold more + cells after 5 days) hence changing the commitment of some cells. Most pubic mRNA data in CD4+T-cells in RA and HC do not use activated cells and as such, the full extent of the effect of DNA-methylation changes cannot be seen, while difference may still suggest some increased basal expression under a resting state. Tissue culture experiments would be indeed necessary to demonstrate a direct effect on TNF or any of the other genes, while not feasible within the 10 days allocated for the revision.  For MIR21, the literature in RA is abundant and related to expression in the serum but not in individual cell types, with increased levels of MIR21 expression in RA patients.

The initial added paragraph was modified. As this is not related to the main message of the study, we did not provide as much details as in our response, hoping this will be satisfactory.

The selection of the TNF gene as the best candidate for the design of a qMSP assay was imposed by the constraints of the assay design but it also is high relevance to RA pathology. We previously demonstrated that the promoter of this gene is differentially methylated in RA (using both 450K array and direct bisulfite sequencing) early in the disease process(39) and used public transcriptomic data in total CD4+T-cells (between HC and RA) and immunoassays (performed in house on HC/RA sera) to shown that de-methylation of the DNA in early RA could be indirectly associated with higher expression (mRNA levels) and higher protein released in the serum (ELISA)(39). However, to prove causality between changes in DNA-methylation and changes at the transcriptional level, transcriptomic data using blood derived cells are not the best as cells are in a resting state (39). Cells (notably immune cells) need to be “activated” with appropriate signals to initiate the transcription of genes. As such, experiment using blood CD4+T-cells showed no (low levels of) detectable basal expression of cytokines by intracellular flow cytometry (IL2, IL10, IFN, TNF or IL17) between HC and RA, while following activation, higher % of positive cells were indeed observed in RA(45, + 2 new references). Polarising cells (towards Th17 for example), would further increase the frequency of committed cells, while not affecting the levels of expression in individual cells(45). qMSP assays will therefore allow monitoring of the time when epigenetic commitment events occur in cells, with important utility for identifying the timing of such events over the pre-clinical phases of RA notably.

Reviewer 2 Report

The authors answered the questions satisfactorily, however typographical errors must still be reviewed. For example: female fender (Table 1).

Typographical errors must still be reviewed. For example: female fender (Table 1).

Author Response

The authors answered the questions satisfactorily, however typographical errors must still be reviewed.

We have requested help from a native English speaker senior colleague and he has made some minor changes all over the manuscript.